# Differences in Disability Perception in Systemic Sclerosis: A Mirror Survey of Patients and Health Care Providers

**DOI:** 10.3390/jcm12041359

**Published:** 2023-02-08

**Authors:** Quentin Kirren, Camille Daste, Frantz Foissac, Hendy Abdoul, Sophie Alami, Marie-Eve Carrier, Linda Kwakkenbos, Marie-Martine Lefèvre-Colau, François Rannou, Agathe Papelard, Alexandra Roren, Brett D. Thombs, Luc Mouthon, Christelle Nguyen

**Affiliations:** 1UFR de Médecine, Faculté de Santé, Université Paris Cité, 75006 Paris, France; 2Service de Rééducation et de Réadaptation de l’Appareil Locomoteur et des Pathologies du Rachis, Hôpital Cochin, AP-HP, Centre-Université Paris Cité, 75014 Paris, France; 3INSERM UMR-S 1153, Centre de Recherche Épidémiologie et Statistique Sorbonne Paris Cité, Université Paris Cité, 75004 Paris, France; 4Unité de Recherche Clinique—Centre d’Investigation Clinique Paris Descartes Necker/Cochin, Hôpital Tarnier, 75014 Paris, France; 5EA 7323, Évaluation des Thérapeutiques et Pharmacologie Périnatale et Pédiatrique, Université Paris Cité, 75014 Paris, France; 6Cabinet d’Études Sociologiques Interlis, 75006 Paris, France; 7Lady Davis Institute for Medical Research, Jewish General Hospital, Montreal, QC H3T 1E2, Canada; 8Behavioural Science Institute, Clinical Psychology, Radboud University Nijmegen, 6525 XZ Nijmegen, The Netherlands; 9INSERM UMR-S 1124, Toxicité Environnementale, Cibles Thérapeutiques, Signalisation Cellulaire et Biomarqueurs (T3S), Campus Saint-Germain-des-Prés, Université Paris Cité, 75006 Paris, France; 10Département Universitaire des Sciences de la Rééducation et de la Réadaptation, Faculté de Santé, Université Paris Cité, 75006 Paris, France; 11Departments of Psychiatry, McGill University, Montreal, QC H3A 1G1, Canada; 12Departments of Epidemiology, Biostatistics, and Occupational Health, McGill University, Montreal, QC H3A 1G1, Canada; 13Departments of Medicine, McGill University, Montreal, QC H3A 1G1, Canada; 14Departments of Psychology, McGill University, Montreal, QC H3A 1G1, Canada; 15Departments of Educational and Counselling Psychology, McGill University, Montreal, QC H3A 1G1, Canada; 16Biomedical Ethics Unit, McGill University, Montreal, QC H3A 1G1, Canada; 17Service de Médecine Interne, Hôpital Cochin, AP-HP, Centre-Université de Paris, 75014 Paris, France

**Keywords:** systemic sclerosis, disability, activity limitations, patient-reported outcomes

## Abstract

Differences in disability perception between patients and care providers may impact outcomes. We aimed to explore differences in disability perception between patients and care providers in systemic sclerosis (SSc). We conducted a cross-sectional internet-based mirror survey. SSc patients participating in the online SPIN Cohort and care providers affiliated with 15 scientific societies were surveyed using the Cochin Scleroderma International Classification of Functioning, Disability and Health (ICF)-65 questionnaire, including 65 items (from 0 to 10), representing 9 domains of disability. Mean differences between patients and care providers were calculated. Care providers’ characteristics associated with a mean difference ≥ 2 of 10 points were assessed in multivariate analysis. Answers were analyzed for 109 patients and 105 care providers. The mean age of patients was 55.9 (14.7) years and the disease duration was 10.1 (7.5) years. For all domains of the ICF-65, care providers’ rates were higher than those of patients. The mean difference was 2.4 (1.0) of 10 points. Care providers’ characteristics associated with this difference were organ-based specialty (OR = 7.0 [2.3–21.2]), younger age (OR = 2.7 [1.0–7.1]) and following patients with disease duration ≥5 years (OR = 3.0 [1.1–8.7]). We found systematic differences in disability perception between patients and care providers in SSc.

## 1. Introduction

Systemic sclerosis (SSc) is an autoimmune disease of the connective tissue characterized by the accumulation of collagen in the skin, vessels, and internal organs [1]. SSc is a disabling condition, leading to impairments, activity limitations and participation restrictions [2,3,4,5,6,7,8,9,10,11], as defined by the International Classification of Functioning, Disability and Health (ICF) [12].

Differences in assessments of health status have been reported between patients and care providers in chronic inflammatory rheumatisms [13,14,15,16]. They involve disease activity and other dimensions such as disability and quality of life [17,18,19]. These differences may lead to lower treatment adherence, physician disappointment and patient dissatisfaction [20]. Differences in assessments may in part occur because patients’ self-assessment is more likely associated with subjective symptoms, while care providers’ assessment is more likely associated with readily observable findings [19]. Differences can also affect the perception of a disease [21]. The perception of a disease corresponds to the cognitive representation that someone (e.g., a patient or a care provider) has about a condition [22]. Little is known about the magnitude of differences in disability perception in SSc between patients and care providers.

Because activity limitations and participation restrictions are complex dimensions of human functioning, including both objective and subjective symptoms, we aimed to describe the differences in the perception of activity limitations and participation restrictions in SSc between patients and care providers and to determine which care providers’ baseline characteristics were associated with these differences.

## 2. Materials and Methods

Design. Difference in perception is usually assessed in mirror studies that compare the responses to the same questionnaire between 2 study populations [22]. Therefore, we conducted a cross-sectional internet-based mirror survey of French-speaking patients with SSc and care providers. The present study is reported in accordance with the Checklist for Reporting Results of Internet E-surveys (Appendix A) [23]. A minimal ratio of 1:1 participants per item was needed (n = 65) [22]. The study was conducted in accordance with the Declaration of Helsinki, and approved by the Comité d’Éthique de la Recherche AP-HP.5 (IRB registration: #00011928) (protocol code 2020-01-05, 20 April 2020). Informed consent was obtained from all subjects involved in the study. All methods were performed in accordance with the relevant guidelines and regulations.

Survey instrument. In a previous study (SCISCIF I), we developed a comprehensive ICF core set for SSc. The self-administered Cochin Scleroderma ICF-65 questionnaire was derived from the “activity and participation” section of the comprehensive ICF core set and includes 9 domains [24] (Appendix A). Each item is scored on an 11-point numerical rating scale from 0 (no limitation) to 10 (maximal limitation). In a second study (SCISCIF II), this questionnaire was distributed to patients with SSc to validate a measure instrument of “limitation of activity” and “participation restriction” in SSc [24]. An adapted version of this questionnaire was developed: the acceptability and understandability of this version were preliminarily tested on 10 care providers from the Cochin Hospital department of physical and rehabilitation medicine, who were not included in the final survey. The wording of the questions was optimized according to their feedback. The introductory sentence to the questionnaire was adapted to care providers as follows: “*Thinking about all your patients with SSc, whatever their disease subset or duration, in average, how would you rate their disability (activity limitation/participation restriction) for the following activity?*”. The answer “*I don’t know*” was added to the 11-point numerical rating scale for each item. In addition, care providers’ socio-demographic data and experience in treating people with SSc were collected (Table 1). In the SCISCIF II study, patients with SSc were surveyed online from February to March 2018 using the Cochin Scleroderma ICF-65 questionnaire [24]. The adapted version of the questionnaire was submitted online to care providers from April to September 2019 (Google Form, declaration of conformity to the CNIL: MR3 research in health domain).

Participants. Assessment of patients’ disability perception in SSc has been fully reported in the SCISCIF II study [24]: patients with SSc were recruited among those followed-up in the Cochin Hospital department of internal medicine and participating in the online Scleroderma Patient-centered Intervention Network Cohort [25]. Inclusion criteria for patients were ≥18 years of age and being classified as having SSc according to the 2013 American College of Rheumatology criteria by a physician expert in SSc. To assess the care providers’ disability perception in SSc, the population sample was recruited through the networks of 15 French scientific societies (see acknowledgments) and among care providers of Cochin Hospital departments of physical and rehabilitation medicine and rheumatology. Using the snowball sampling principle, care providers could nominate, through their networks, other participants. The inclusion criterion for care providers was involvement in the management of patients with SSc (e.g., physicians, dentists, residents, nurses, nursing assistants, physiotherapists, occupational therapists, and podiatrists). The exclusion criterion was involvement in the design of the present study and non-French-speaking.

Statistical analyses. Preplanned statistical analyses were performed with R 3.3.1. Items of the questionnaire with more than 20% missing data in patients, and care providers who did not answer at least 50% of the questionnaire or answered more than 20% of the items with “*I don’t know*” were excluded from analyses. Domains of the questionnaire that did not include at least 3 analyzable items were excluded from analyses.

For descriptive analyses, quantitative variables were described by their means and standard deviations, and qualitative variables were described by their absolute and relative frequencies. Differences in the perception between patients and care providers were calculated individually for each care provider, for both the whole questionnaire and the domain levels of the questionnaire. At the item level, the absolute difference (from 0 to 10) for each care provider was calculated as the difference between her or his answer and the mean answer of all patients. At the whole or domain questionnaire level, the overall difference for one care provider was calculated by averaging the differences between the items constituting the domain or the entire questionnaire. We considered that a care provider had a meaningful difference with the patients in disability perception regarding the domains or the entire questionnaire if the individual overall difference was ≥2 of 10 points. This 2-point difference has been used in other studies reporting differences in assessments between patients and care providers [26]. The results of differences in disability perception for all care providers were presented as a descriptive statistic of the individual quantitative or qualitative differences.

For comparative analyses, qualitative variables were compared using Fisher’s exact test between care providers who had a mean difference ≥ 2 of 10 points for the whole questionnaire and those who did not. To identify care providers’ baseline variables associated with the probability of differences in disability perception, we conducted a binary logistic regression. Four baseline variables were prespecified, based on our clinical experience: (i) organ-based specialty (i.e., dermatologists, cardiologists, pneumologists and rheumatologists), (ii) following patients with mean disease duration ≥ 5 years (reference < 5 years), (iii) following < 50 patients a year for DcSSc and LcSSc versus following ≥ 50 patients for at least one disease condition or ≥50 patients for both conditions and (iv) age < 45 years (reference ≥ 45 years). We presented the complete logistic model. For all analyses, a *p*-value < 0.05 was considered statistically significant. We did not perform post hoc comparisons.

## 3. Results

Response rate. Overall, 109 care providers met the inclusion criteria and participated in our survey. As the survey was submitted by newsletters from scientific societies, it was not possible to evaluate the number of people approached and to establish a response rate. Four care providers were excluded from analyses because they answered less than 50% of the questionnaire. In total, 53/105 (51%) included care providers who were women and 34/105 (32%) were between 35 and 44 years old, and there were 75/105 (71%) and 92/105 (88%) care providers who followed less than 50 patients with LcSSc or DcSSc a year, respectively (Table 1). Overall, 184 patients were approached in the SCISCIF II study, 113 patients (61%) participated and 109 (60%) completed the survey [27]. The mean age was 55.9 (14.7) years, the mean disease duration was 10.1 (7.5) years, 57/109 (52%) had LcSSc and 41 (38%) had DcSSc (Table 2). Overall, 12 items of the questionnaire were excluded from analyses: 2 items had more than 20% missing data in patients (items 58 and 59), and 10 items had more than 20% of care providers answering “*I don’t know*” (items 47, 58, 49, 50, 51, 53, 54, 55, 64 and 65) (see details in Appendix A). Two domains of the questionnaire (interpersonal interactions and relationships, major life areas) were not analyzed because they did not include at least three analyzable items.

Participants. Characteristics of the patient population in SCISCIF II [27], regarding subset, mean age, sex and functional impact with mean Health Assessment Questionnaire (HAQ), were similar to other large cohorts in the medical literature except that there was a higher proportion of sine scleroderma subset in our population [27]. The population of care providers was representative of the variety of specialties involved in the treatment of this disease. However, the proportion of specialties was not well balanced with a majority of internal medicine physicians corresponding to the primary care provider involved in the management of SSc, especially forms with a greater functional impact.

Differences in disability perception between patients and care providers. The mean (SD) for ratings for the whole questionnaire was 2.4 (0.8) of 10 points for patients and 4.4 (1.4) of 10 points for care providers. For all analyzable domains of the questionnaire, care providers’ rates were numerically higher than those of patients. The mean difference between patients and care providers’ answers was 2.4 (1.0) of 10 points, and 64/105 (61%) care providers had a mean difference of ≥ 2 of 10 points for the whole questionnaire. The domain for which the magnitude of differences in ratings between patients and care providers was the largest was “self-care” (2.9 [1.5] of 10 points), and the domain for which it was the smallest was “learning and applying knowledge” (2.1 [1.0] of 10 points) (Table 3).

Care providers’ baseline variables associated with differences in disability perception. In the univariate analysis, the percentage of organ-based specialists was significantly larger in the group of care providers who had a mean difference ≥ 2 of 10 points for the whole questionnaire than in the group of care providers who did not (6/41 [15%] versus 30/64 [47%], *p* < 0.01) (Table 4). In the multivariate analysis, care providers’ baseline variables associated with a mean difference ≥ 2 of 10 points for the whole questionnaire were organ-based specialty (OR = 7.0, 95% CI [2.3 to 21.2], *p* < 0.001), younger age (OR = 2.7, 95% CI [1.0 to 7.1], *p* = 0.04), and following patients with disease duration ≥ 5 years (OR = 3.0, 95% CI [1.1 to 8.7], *p* = 0.04) (Table 5).

## 4. Discussion

In the present study, care providers’ rates were systematically higher than those of patients for all analyzable domains of the ICF-65 questionnaire. The majority of them (61%) had a meaningful difference (≥2 of 10 points) for the whole questionnaire.

Care providers may have misperceived disability because their disability perception was care-provider-centered rather than patient-centered and/or because the patients they follow were different from those surveyed in the present study. In the context of a rare disease, they may also have lacked experience: more than 70% of care providers reported following less than 50 patients with DSSc per year, and 10/65 items of the Cochin Scleroderma ICF-65 questionnaire had more than 20% of care providers answering “*I don’t know*”. Another plausible reason is that SSc patients included in the present study had long disease duration. Therefore, a response shift may have contributed to greater resilience regarding activity limitations and participation restrictions in our patients than in those with shorter disease duration and to the successful implementation of environmental adaptative solutions to reduce the burden of SSc on their functioning [28]. These aspects may be difficult to capture for care providers.

The domain for which the magnitude of differences in ratings between patients and care providers was the largest was “self-care” and the domain for which it was the smallest was “learning and applying knowledge”. These results were unexpected because “self-care” is usually more easily accessible to assessments, using interviewer-, proxy- or self-administered questionnaires, than “learning and applying knowledge”. In chronic obstructive pulmonary disease, a mirror study reported larger differences in perception between patients and physicians for leisure, social life and sexual intercourse than for work and activities of daily living [29]. A reason might be that care providers overestimated domains in which they did not feel comfortable with clarifying their disability perception.

In multivariate analysis, care providers’ characteristics associated with the difference in perception were an organ-based specialty, younger age and following patients with disease duration ≥5 years. Arat and colleagues reported differences in the perception of patients with SSc between general practitioners and organ-based specialists [30], between organ-based specialists [31]. They also reported that illness perceptions may influence the ability of patients with SSc to cope with health problems [32]. The precise causes of such differences remain unclear. It could be the case that organ-based specialists have a vision of the disease more focused on impairments rather than activity and participation, or they have recruitment of patients with more specific organ impairments and lower overall impact on functioning than non-organ-based specialists, or both. Arat and colleagues also reported that work experience was associated with differences in functioning perception [30]. We chose a cut-off of 45 years to define the younger age of care providers because it reflects work experience of less than 10 years. Interestingly, we found that all 5/105 (4.8%) care providers who reported following more than 150 patients with SSc a year did not have a meaningful difference in perception, suggesting that practitioners’ experience may enhance functioning perception. Concerning disease duration, one can hypothesize that determinants of disability become more complex and difficult to capture for care providers in patients with longer disease duration, that patients with longer disease duration have greater resilience, or both [33,34,35].

Our study has limitations. Web-based surveying exposes selection biases such as self-selection (i.e., “voluntary effect”: the most experienced patients and care providers are more likely to participate in the survey) [23]. All participants were followed in a referral center for SSc patients and may not be representative of all patients with SSc in France, and not comparable to the patients followed by all the care providers who participated in the study. The best comparison would be having the participants assessing themselves, and their main “provider reference” assessing the corresponding participants, consistently, and because providers were from different hospitals and care facilities, the same should have been undertaken for patients from different parts of France and followed by different specialties. Depending on countries and specific contexts, rheumatology may not be considered an organ-based specialty. Modifying the groups for this variable may have yielded different results. Finally, the 45 years of age corresponding to 10 years of experience was speculative, and “years of experience with SSc care” would have been a more appropriate candidate for the multivariate regression.

In summary, differences in disability perception in SSc between patients and care providers exist. Care providers’ baseline characteristics associated with these differences were an organ-based specialty, younger age and following patients with disease duration ≥ 5 years. Our results further support the use of assessment tools centered on patients’ perspectives to complement available assessment tools centered on disease, in order to better capture patients’ priorities [27,36]. In the absence of SSc curative therapies, bridging the gap in disability perception between patients and care providers is vital to design personalized multidisciplinary interventions targeting more specifically patient-important outcomes [37].

## Figures and Tables

**Table 1 jcm-12-01359-t001:** Care providers’ self-reported characteristics (n = 105).

Care Providers’ Characteristics	n (%)
Age	
• <25 years	3 (3)
• 25–34 years	20 (19)
• 35–44 years	34 (32)
• 45–54 years	22 (21)
• 55–64 years	22 (21)
• ≥65 years	4 (4)
Gender	
• Women	53 (50)
• Men	52 (50)
Degree	
• Graduated > 2 years	87 (83)
• Graduated ≤ 2 years	10 (10)
• Not graduated	8 (7)
Occupation	
• Board-certified physicians	82 (78)
• Medical residents	9 (8)
• Physiotherapists	7 (7)
• Nurses	4 (4)
• Dental surgeons	1 (1)
• Occupational therapists	1 (1)
• Nursing assistants	1 (1)
Medical specialties	
• Internal medicine	57 (54)
• Rheumatology	20 (19)
• Pulmonology	11 (11)
• Dermatology	4 (4)
• Physical and rehabilitation medicine	3 (3)
• Other	3 (3)
• Cardiovascular medicine	1 (1)
Predominant setting for practice	
• Hospital	60 (57)
• Hospital and academia	24 (23)
• Academia	14 (13)
• Private practice	4 (4)
• Hospital and private practice	1 (1)
• Academia and private practice	1 (1)
• Other	1 (1)
Number of patients with limited cutaneous systemic sclerosis followed-up per year	
• <50 patients	75 (71)
• 50–149 patients	20 (19)
• ≥150 patients	5 (5)
• Not applicable	5 (5)
Number of patients with diffuse cutaneous systemic sclerosis followed-up per year	
• <50 patients	92 (88)
• 50–149 patients	10 (9)
• ≥150 patients	1 (1)
• Not applicable	2 (2)
Proportion of female patients followed-up per year	
• <25%	5 (5)
• 25–50%	6 (6)
• 50–75%	45 (43)
• >75%	49 (46)
Patients’ mean disease duration	
• <1 year	0 (0)
• 1–5 years	33 (31)
• 5–10 years	63 (60)
• >10 years	9 (9)
Patients’ mean age	
• <25 years	0 (0)
• 25–34 years	5 (5)
• 35–44 years	20 (19)
• 45–54 years	40 (38)
• 55–64 years	37 (35)
• ≥65 years	3 (3)

**Table 2 jcm-12-01359-t002:** Patients’ characteristics (n = 109).

Characteristics	
Women, n/N (%)	98/109 (90)
Age (years), mean (SD)	55.9 (14.7) ^a^
Body mass index (kg/m^2^), mean (SD)	23.0 (4.7) ^a^
Disease duration (years), mean (SD)	10.1 (7.5) ^a^
Systemic sclerosis subset, n/N (%)	
• Limited cutaneous	57/109 (52)
• Diffuse cutaneous	41/109 (38)
• Sine scleroderma	8/109 (7)
• Unspecified	3/109 (3)
Impairments, n/N (%)	
• Modified Rodnan skin score (0 to 51), mean (SD)	7.7 (8.4) ^b^
• Sclerodactylia, n/N (%)	81/106 (76)
• Telangectasias, n/N (%)	66/107 (62)
• Pulmonary fibrosis, n/N (%)	45/107 (42)
• Digital ulcer, n/N (%)	44/107 (41)
• Stiffness of small joints (fingers, wrists), n/N (%)	35/104 (34)
• Gastrointestinal tract involvement, n/N (%)	31/106 (29)
• Stiffness of large joints (elbows, hips, knees, ankles), n/N (%)	21/103 (20)
• Pulmonary arterial hypertension, n/N (%)	7/107 (7)
• Scleroderma renal crisis, n/N (%)	6/107 (6)
Activity limitations scores, mean (SD)	
• Health Assessment Questionnaire (0 to 3), mean (SD)	1.1 (0.8)
• Scleroderma Health Assessment Questionnaire (0 to 3), mean (SD)	1.0 (0.7)
• Cochin Hand Function scale (0 to 90), mean (SD)	18.0 (18.7)
• Mouth Handicap in Systemic Sclerosis scale (0 to 48), mean (SD)	19.0 (12.5)

^a^ n = 107; ^b^ n = 104.

**Table 3 jcm-12-01359-t003:** Differences in disability perception between patients and care providers for each of the 9 domains of the Cochin Scleroderma ICF-65 questionnaire.

	Mean (SD) Score in Patients (0–10)	Mean (SD) Score in Care Providers (0–10)	Mean (SD) Absolute Difference between Patients and Care Providers (0–10)	Care Providers with Mean Absolute Difference ≥2 of 10 Points, n/N (%)
Cochin Scleroderma ICF-65 whole questionnaire (65 items)	2.4 (0.8)	4.4 (1.4)	2.4 (1.0)	64/105 (61)
Cochin Scleroderma ICF-65 domains				
• Self-care (6 items)	1.6 (0.3)	4.4 (1.8)	2.9 (1.5)	72/105 (68)
• Domestic life (5 items)	2.7 (0.4)	5.4 (1.8)	2.8 (1.5)	72/105 (68)
• Mobility (18 items)	2.7 (0.9)	4.8 (1.5)	2.4 (1.1)	62/105 (59)
• Community, social and civic life (6 items)	3.3 (1.1)	5.4 (1.7)	2.4 (1.2)	61/105 (55)
• Communication (4 items)	1.2 (0.1)	2.8 (1.6)	2.2 (1.2)	55/105 (52)
• General tasks and demands (8 items)	2.6 (0.5)	4.2 (1.8)	2.2 (1.2)	55/105 (52)
• Learning and applying knowledge (4 items)	2.1 (0.5)	3.5 (1.7)	2.1 (1.0)	53/105 (46)
• Interpersonal interactions and relationships (9 items)	2.6 (0.2) *	2.0 (1.7) *	Not analyzable	Not analyzable
• Major life areas (5 items)	2.6 (0.2) **	4.9 (2.0) **	Not analyzable	Not analyzable

Scores calculated based on the answers to * 2/9 items and ** 2/5 items, respectively (other items excluded because more than 20% missing data in patients or more than 20% of care providers answering “*I don’t know*”).

**Table 4 jcm-12-01359-t004:** Care providers’ baseline variables associated with differences in disability perception: results of the univariate analysis.

N (%)	Care Providers with Mean Absolute Difference < 2/10 (n = 41)	Care Providers with Mean Absolute Difference ≥ 2/10 (n = 64)	*p*-Value *
Age ≥ 45 years	22 (54)	26 (41)	0.23
Women	18 (44)	35 (55)	0.32
Graduated > 2 years	36 (88)	51 (80)	0.62
Organ-based specialty	6 (15)	30 (47)	<0.01
≥50 patients with DcSSc followed-up a year	5 (12)	6 (9)	0.88
≥50 patients with LcSSc followed-up a year	14 (34)	11 (17)	0.12
Patients with mean disease duration ≥ 5 years	26 (63)	46 (72)	0.39

* Comparisons between groups were performed using Fisher’s exact test. A *p*-value < 0.05 was considered statistically significant. DcSSc: diffuse cutaneous systemic sclerosis; LcSSc: limited cutaneous systemic sclerosis.

**Table 5 jcm-12-01359-t005:** Care providers’ baseline variables associated with probability of difference in disability perception: results of the multivariate logistic analysis.

Care Providers’ Characteristics	OR	95% Confidence Interval	*p*-Value *
Organ-based specialty	7.0	2.3–21.2	<0.001
Following patients with mean disease duration ≥ 5 years	3.0	1.1–8.7	0.04
≥50 patients with DcSSc or LcSSc followed-up a year versus < 50 patients	0.4	0.1–1.5	0.18
≥50 patients with DcSSc and LcSSc followed-up a year versus < 50 patients	0.4	0.1–2.0	0.27
Age < 45 years	2.7	1.0–7.1	0.04

* A *p*-value < 0.05 was considered statistically significant. DcSSc: diffuse cutaneous systemic sclerosis; LcSSc: limited cutaneous systemic sclerosis; OR: odds ratio.

## Data Availability

Full original protocol and dataset can be accessed upon request for academic researchers by contacting Christelle Nguyen (christelle.nguyen2@aphp.fr). Statistical codes can be accessed upon request for academic researchers by contacting Frantz Foissac (frantz.foissac@aphp.fr).

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
