# Peer review of "Differences in Disability Perception in Systemic Sclerosis: A Mirror Survey of Patients and Health Care Providers"

_jcm, 2023, doi:10.3390/jcm12041359_

Round 1
Reviewer 1 Report
Authors belonging to a scleroderma expert center contribute to a rarely explored but important item in rheumatic diseases.
Methodology is sound and results adequately discussed. This might contribute to develop better care for this severe disease in the futur.
Only one remark for completeness. Authors might read and include results in introduction or discussion of a paper on a large cohort from the Belgian sclero registry: Arat S et al The association of illness perceptions with physical and mental health in systemic sclerosis patients: an exploratory study. Musculoskeletal Care. 2012 Mar;10(1):18-28.
Author Response
Authors belonging to a scleroderma expert center contribute to a rarely explored but important item in rheumatic diseases.
Methodology is sound and results adequately discussed. This might contribute to develop better care for this severe disease in the futur.
Only one remark for completeness. Authors might read and include results in introduction or discussion of a paper on a large cohort from the Belgian sclero registry: Arat S et al The association of illness perceptions with physical and mental health in systemic sclerosis patients: an exploratory study. Musculoskeletal Care. 2012 Mar;10(1):18-28.
Thank you. We now mentioned the results of this paper in the discussion (ref 32, page 8/11).
Reviewer 2 Report
I read with interest the paper from Kirren et al, regarding the different perception of disability between patients and care providers.
The manuscript is somewhat novel in the topic, not considering care givers (people daily helping at home), but mostly people who are involved in care but not in daily life (mostly doctors).
The study has some methodological weakness, which cannot be overcome at this stage only partially. The rest shold be acknowledged as limitation.
1. Patients evaluate themselves, providers evaluate the SSc population they take care of. As acknowledged, the average of my SSc cohort as a provider, might be very different from the average of a single hospital. The study could be re-run having the patients assessing themselves and their main "provider reference" assessing that patient. That would make the best comparison.
2. As there were providers from different hospital and care facilities, the same should have been for patients, from different part of France and followed by different specialties.
3. I might be personally biased given my background and the French situation might be different, but Rheumatology is not an organ-based specialty. Rheumatologist take care of all SSc aspects as Internists do. I would therefore modify the groups for this variable and repeat the analysis.
4. The 45 years of age corresponding to 10 years of experience is very speculative and "years of experience with SSc care" would have been a much more appropriate candidate for the multivariate regression. If you are trained in an SSc referral center, you will see more SSc patients during your residency than a 60 yo doctor in his whole private practice working life. I would consider Age as a continuous variable, to make it less artificial, but still this will not really reflect "experience with SSc".
Author Response
I read with interest the paper from Kirren et al, regarding the different perception of disability between patients and care providers.
The manuscript is somewhat novel in the topic, not considering care givers (people daily helping at home), but mostly people who are involved in care but not in daily life (mostly doctors).
The study has some methodological weakness, which cannot be overcome at this stage only partially. The rest shold be acknowledged as limitation.
- Patients evaluate themselves, providers evaluate the SSc population they take care of. As acknowledged, the average of my SSc cohort as a provider, might be very different from the average of a single hospital. The study could be re-run having the patients assessing themselves and their main "provider reference" assessing that patient. That would make the best comparison.
Thank you. We now mentioned this important point as a limitation of the present study (page 8/11).
- As there were providers from different hospital and care facilities, the same should have been for patients, from different part of France and followed by different specialties.
Thank you. We now mentioned this point as a limitation of the present study (page 8/11).
- I might be personally biased given my background and the French situation might be different, but Rheumatology is not an organ-based specialty. Rheumatologist take care of all SSc aspects as Internists do. I would therefore modify the groups for this variable and repeat the analysis.
Thank you. However, because hypotheses were prespecified in the protocol, we do not believe it would be methodologically sound to rerun analyses a posteriori, after the results were already obtained. However, we mentioned this point as a perspective in the discussion (page 8/11).
- The 45 years of age corresponding to 10 years of experience is very speculative and "years of experience with SSc care" would have been a much more appropriate candidate for the multivariate regression. If you are trained in an SSc referral center, you will see more SSc patients during your residency than a 60 yo doctor in his whole private practice working life. I would consider Age as a continuous variable, to make it less artificial, but still this will not really reflect "experience with SSc".
Thank you. Again, because hypotheses were formulated a priori, we do not believe it would be methodologically sound to rerun analyses a posteriori, after the results were already obtained. However, we mentioned this point as a perspective in the discussion (pages 8 and 9/11).
Round 2
Reviewer 2 Report
Thanks for addressing my comments, no further request from my side.